# Dynamic Characteristics of the Bouc–Wen Nonlinear Isolation System

**Zhiying Zhang, Xin Tian and Xin Ge ***

School of Human Settlements and Civil Engineering, Xi'an Jiaotong University, Xi'an 710049, China; zhangzhy@mail.xjtu.edu.cn (Z.Z.); XJTUTIANXIN@163.com (X.T.)
* Correspondence: XJTUGEXIN@163.com

**Abstract:** The Bouc–Wen nonlinear hysteretic model has many control parameters, which has been widely used in the field of seismic isolation. The isolation layer is the most important part of the isolation system, which can be effectively simulated by the Bouc–Wen model, and the isolation system can reflect different dynamic characteristics under different control parameters. Therefore, this paper mainly studies and analyzes the nonlinear dynamic characteristics of the isolation system under different influence factors based on the incremental harmonic balance method, which can provide the basis for the dynamic design of the isolation system.

**Keywords:** Bouc–Wen; nonlinear hysteretic model; incremental harmonic balance method; vibration isolation system

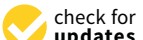



## 1. Introduction

In recent years, the idea of seismic isolation building is widely used in civil engineering, vehicle engineering, aerospace engineering and other fields, such as the construction of commercial complexes, hospitals, residential buildings, tunnels and subway stations, shock absorption of vehicles and launch of rockets, etc. This is achieved mainly through the installation of seismic energy-absorption devices or coating with damping materials to achieve the optimal application of different material characteristics, which can ensure that the isolation system effectively protect the superstructure from earthquake damage or damage. At present, with the continuous development of seismic isolation technology, countries around the world are also carrying out research on seismic isolation theoretical systems related to seismic isolation structures [1–14].

As one of the most important parts of the isolation structure, the mechanical properties of the isolation layer are mainly hysteretic characteristics; meanwhile, the hysteretic nonlinearity system usually is a kind of multi-valued, non-analytical system, so it is very difficult to establish a mathematical model of the universal hysteretic nonlinear system. In previous works, Caughey used a bilinear hysteretic model to study the random vibration of the system [15], and some scholars also used a piecewise linear hysteretic model to analyze the system response [16]. However, in the piecewise linear model, due to the abrupt change in stiffness, it is difficult to reflect the yield characteristics of the system. For this reason, relevant scholars began to use the smooth curve model to analyze a hysteretic system [17]. In 1967, Bouc proposed a simple and smooth hysteretic model controlled by differential equations. Later, Wen et al. improved this model and proved that this model could generate a series of different hysteretic loops [18]. Subsequently, the Bouc–Wen model was developed as a smooth curve hysteresis model, which includes both nonlinear stiffness and damping; it can approximate all kinds of smooth hysteretic curves and reflect the mechanical properties of the system under different control parameters well. Therefore, the model has been widely used in research and analysis in the field of engineering applications.

Domestic and foreign scholars have also made relevant explorations regarding the solving methods of nonlinear systems. For example, Marano used the stochastic param-

eter analysis method to describe the response of the seismic isolation bridge model and evaluated the performance of the seismic isolation device [19]. Meibodi used Incremental Dynamic Analysis (IDA) to determine the response of analyzing the nonlinear model [20]. TSiatas adopted the state space method to analyze the Hysteretic Energy Sink [21]. Li used a numerical simulation method to mainly analyze the parameter characteristics of a nonlinear hysteretic system [22]. Zhu adopted the normalized Bouc–Wen model to describe the hysteretic characteristics of a magnetorheological damper [23]. Kim analyzed the asymmetric hysteresis behavior of the Bouc–Wen model based on phenomenology [24]. Zhu established the generalized Bouc–Wen model to accurately describe the nonlinear phenomenon of the piezoelectric actuator, and then compensated the hysteresis effect to solve the problem [25,26]. Niola proposed a Traceless Kalman Filter to identify the parameters of the hysteretic model, and the effectiveness of this method was verified by numerical simulation and experiments [27]. Casalotti used the asymptotic analysis method and path-tracking method to study the resonance dynamics behavior of nonlinear structures [28]. Wu analyzed the modified Bouc–Wen model with MATLAB and established the nonlinear hysteretic suspension model. The frequency response characteristics of a semi-active suspension were studied [29]. Bai used the incremental harmonic balance method (IHB) to analyze the cubic nonlinear viscous damped vibration system, which was verified by the fourth-order Runge–Kutta numerical integration [30]. Hossein used the incremental harmonic balance method to study the dynamic behavior and stability of the two-degree-of-freedom nonlinear system [31]. Liu studied two-degree-of- freedom systems with piecewise linearity and has done many other work in this field [32,33]. At present, the methods to analyze hysteretic systems mainly include the slow variation method or harmonic balance method, etc. However, according to the relevant results of numerical analysis technology at home and abroad, it can be found that the main research is on the frequency response curve and bifurcation diagram of the relevant model, and to distinguish the stable state, whereas research on the control parameter space of the system is obviously insufficient. In this regard, the accurate and effective prediction of system behavior by adjusting the major parameters has gradually become a highly popular topic [34].

## 2. Mechanical Properties of the Bouc–Wen Model

The Bouc–Wen model (the Bouc–Wen nonlinear isolation model with two degrees of freedom) is a hysteretic model widely used in the engineering and scientific research fields at present, and specifically widely used in the range of seismic isolation engineering. Moreover, the hysteretic restoring force and displacement diagram of the model is smooth and continuous, and the restoring force of the seismic isolation structure can be simulated by using the model, as shown in Figure 1.

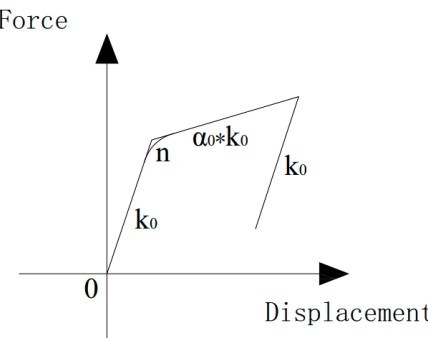

**Figure 1.** Schematic diagram of the Bouc–Wen hysteresis.

The force–displacement relationship of this model is expressed as follows:

$$F(x) = \alpha k_0 x + (1 - \alpha)k_0 z \tag{1}$$

In addition, the following equations, Equations (2) and (3), are satisfied:

$$\dot{z} = \frac{1}{x_y}\left(A\dot{x}(t) - \beta\dot{x}(t)|z^n(t)| - \gamma|\dot{x}(t)|z^n(t)\right) n = 1,3,5\ldots \tag{2}$$

$$\dot{z} = \frac{1}{x_y}\left(A\dot{x}(t) - \beta\dot{x}(t)z^n(t) - \gamma|\dot{x}(t)|\left|z^{n-1}(t)\right||z(t)|\right) n = 2,4,6\ldots \tag{3}$$

where $Z$ is the hysteretic displacement of the system, the area of the hysteretic loop controlled by the parameter $A$; the shape of the hysteretic loop is controlled by parameters $\beta$ and $\gamma$; the smoothness of the hysteretic curve controlled by $n$; and $\alpha$ represents the ratio of the back-to-front stiffness of the yield value. When $\alpha = 1$, the system has ordinary linear elasticity.

For different values of parameters $\beta$ and $\gamma$, the hysteretic characteristics of the structure will change in different forms. When $\beta + \gamma > 0$, as shown in Figure 2a,b, the structure presents soft characteristics, and the hysteretic restoring force of the system decreases as the displacement response increases. When $\beta + \gamma = 0$, the structure is linear in the loading stage. When $\beta + \gamma < 0$, the structure shows hardening properties, and the hysteretic restoring force of the system increases with the increase in displacement response. As we can be seen from Figure 2b,d, when the ratio of $\beta/|\gamma|$ is relatively large, the enveloping graph of the system hysteretic restoring force curve is larger and the curve shape is relatively full, which proves that the structural system consumes more energy under this parameter. When the ratio is small, it can be seen that the overall enveloped area of Figure 2a,c is small, and the energy dissipation characteristics of the structural system are relatively weak compared with the others.

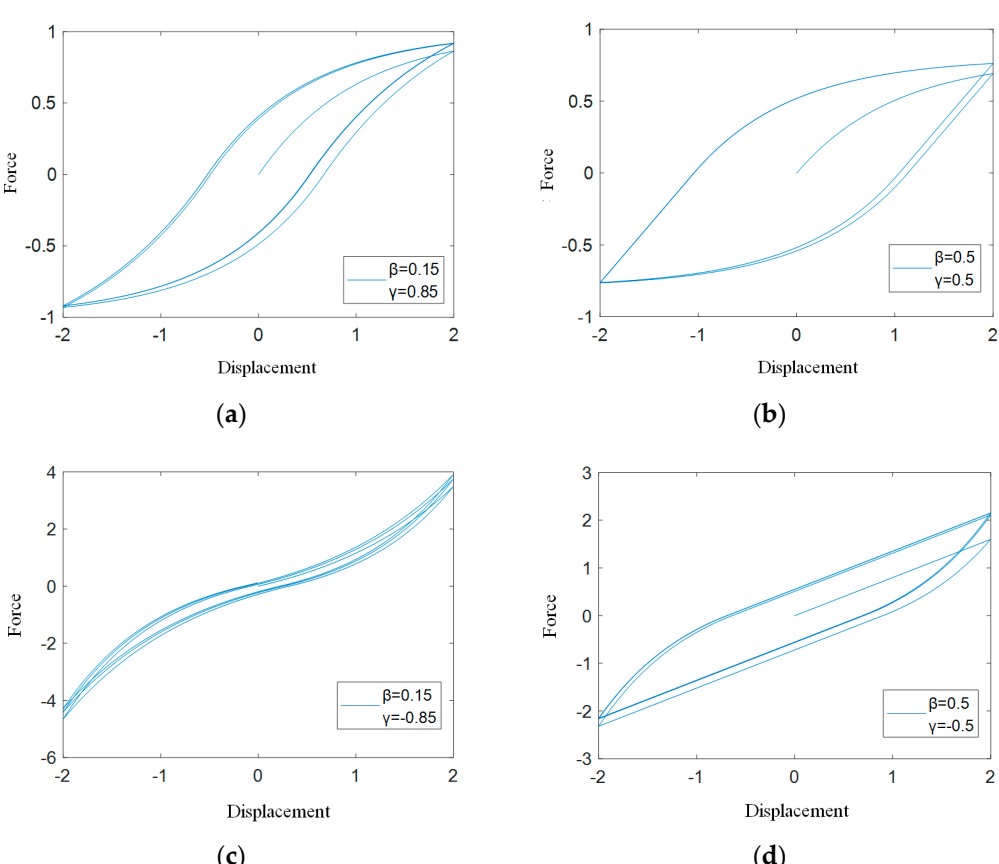

**Figure 2.** Bouc–Wen hysteresis loops with different parameters. (The values of β in (**a–d**) are 0.15, 0.5, 0.15, 0.5; The values of γ in (**a–d**) are 0.85, 0.5, −0.85, −0.5.)

## 3. Theoretical Analysis of the Nonlinear Isolation System

The structure presents strong nonlinear characteristics under the action of an actual strong earthquake. For the isolation structure, the material properties of the isolation layer differ greatly from the structure, so the isolation structure can be divided into the superstructure and the substructure. The characteristics of the isolation layer are coupled and simplified by the Bouc–Wen nonlinear model, in which the horizontal stiffness of the superstructure is large and usually presents a translational state. In order to study the nonlinear dynamic characteristics of the isolation system, a strong nonlinear solution method—Incremental Harmonic Balance Method (IHB) [35] as adopted in this paper, to analyze the Bouc–Wen nonlinear model and to study the influence law of the different control parameters under the steady-state response of the system.

The Bouc–Wen nonlinear isolation model with two degrees of freedom under harmonic excitation is shown in Figure 3a,b.

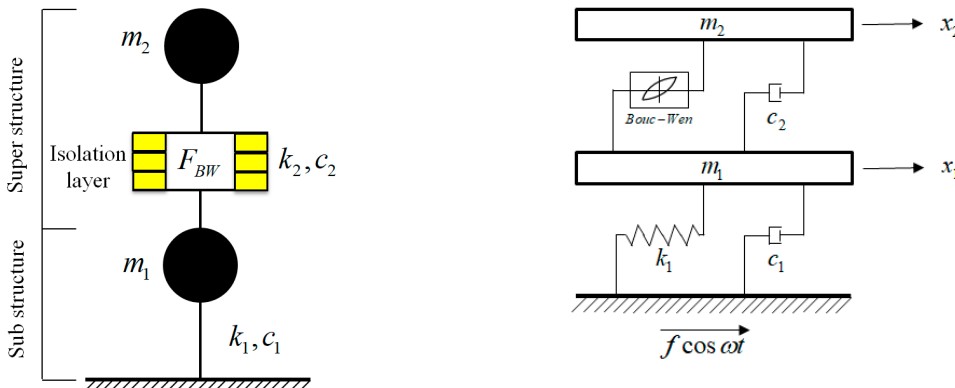

**Figure 3.** Simplified analysis model of the isolation system. (**a**) Simplified model; (**b**) Analysis model.

The dynamic equations of a two-degree-of-freedom Bouc–Wen system are as follows:

$$\begin{cases} m_1\ddot{x}_1 + c_1\dot{x}_1 + k_1 x_1 + c_2(\dot{x}_1 - \dot{x}_2) - F_{BW} = f\cos\omega t \\ m_2\ddot{x}_2 + c_2(\dot{x}_2 - \dot{x}_1) + F_{BW} = 0 \\ \dot{z} = \dot{x}_2 - \dot{x}_1 - \beta|\dot{x}_2 - \dot{x}_1|z - \gamma(\dot{x}_2 - \dot{x}_1)|z| \\ F_{BW} = \alpha k_2(x_2 - x_1) + (1-\alpha)k_2 z \end{cases} \tag{4}$$

Written in matrix form:

$$\begin{bmatrix} m_1 & 0 & 0 \\ 0 & m_2 & 0 \\ 0 & 0 & 0 \end{bmatrix}\begin{bmatrix} \ddot{x}_1 \\ \ddot{x}_2 \\ \ddot{z} \end{bmatrix} + \begin{bmatrix} c_1+c_2 & -c_2 & 0 \\ -c_2 & c_2 & 0 \\ 1 & -1 & 1 \end{bmatrix}\begin{bmatrix} \dot{x}_1 \\ \dot{x}_2 \\ \dot{z} \end{bmatrix} + \begin{bmatrix} \alpha k_2 + k_1 & -\alpha k_2 & -(1-\alpha)k_2 \\ -\alpha k_2 & \alpha k_2 & (1-\alpha)k_2 \\ 0 & 0 & 0 \end{bmatrix}\begin{bmatrix} x_1 \\ x_2 \\ z \end{bmatrix} + $$
$$\begin{bmatrix} 0 \\ 0 \\ \beta|\dot{x}_2 - \dot{x}_1|z + \gamma(\dot{x}_2 - \dot{x}_1)|z| \end{bmatrix} = \begin{bmatrix} f \\ 0 \\ 0 \end{bmatrix}\cos\omega t \tag{5}$$

among which

$$M = \begin{bmatrix} m_1 & 0 & 0 \\ 0 & m_2 & 0 \\ 0 & 0 & 0 \end{bmatrix} \quad C = \begin{bmatrix} c_1+c_2 & -c_2 & 0 \\ -c_2 & c_2 & 0 \\ 1 & -1 & 1 \end{bmatrix} \quad K = \begin{bmatrix} \alpha k_2 + k_1 & -\alpha k_2 & -(1-\alpha)k_2 \\ -\alpha k_2 & \alpha k_2 & (1-\alpha)k_2 \\ 0 & 0 & 0 \end{bmatrix}$$

$$K_n = \begin{bmatrix} K_{n(1)} \\ K_{n(2)} \\ K_{n(3)} \end{bmatrix} = \begin{bmatrix} 0 \\ 0 \\ \beta|\dot{x}_2 - \dot{x}_1|z + \gamma(\dot{x}_2 - \dot{x}_1)|z| \end{bmatrix}$$

where $M$, $C$, $K$ and $K_n$, respectively, represent the structural mass, damping, linear stiffness and nonlinear hysteretic matrix, and introduce the time scale. Then, Equation (5) can be written as

$$\omega^2 M\ddot{x} + \omega C\dot{x} + Kx + \omega K_n = F\cos\tau \tag{6}$$

among which

$$F = \begin{bmatrix} f & 0 & 0 \end{bmatrix}^T x = \begin{bmatrix} x_1 & x_2 & z \end{bmatrix}^T$$

Applying incremental processes:

$$X_{(i)} = X_{(i0)} + \Delta X_{(i)}, \; X_{(1)} = x_1, \; X_{(2)} = x_2, \; X_{(3)} = z, \; \omega = \omega_0 + \Delta\omega \tag{7}$$

where $\Delta X_{(i)}$ and $\Delta\omega$ are weak increments. Substitute Equation (7) into Equation (6), and then ignore the high-order trace, and only retain the items below the first order, as shown in Equation (8).

$$\omega_0{}^2 M\Delta\ddot{x} + \omega_0 C\Delta\dot{x} + K\Delta x + K_{nd}\Delta x + C_{nd}\Delta\dot{x} = \Phi - \left[2\omega_0 M\ddot{x}_0 + C\dot{x}_0 + K_n(\dot{x}_0)\right]\Delta\omega$$
$$\Phi = F\cos(\tau) - \omega_0 K_n(\dot{x}_0) - (\omega_0{}^2 M\ddot{x}_0 + \omega_0 C\dot{x}_0 + Kx_0) \tag{8}$$

where $K_{nd}$ and $C_{nd}$ are the partial derivatives of the nonlinear term $K_n$ with respect to $x$ and $\dot{x}$, which can be written in the Jacobian matrix form as follows:

$$K_{nd} = \frac{\partial K_{n(i)}}{\partial X_{(i)}} = \begin{bmatrix} \frac{\partial K_{n(1)}}{\partial x_1} & \frac{\partial K_{n(1)}}{\partial x_2} & \frac{\partial K_{n(1)}}{\partial z} \\ \frac{\partial K_{n(2)}}{\partial x_1} & \frac{\partial K_{n(2)}}{\partial x_2} & \frac{\partial K_{n(2)}}{\partial z} \\ \frac{\partial K_{n(3)}}{\partial x_1} & \frac{\partial K_{n(3)}}{\partial x_2} & \frac{\partial K_{n(3)}}{\partial z} \end{bmatrix} = \begin{bmatrix} 0 & 0 & 0 \\ 0 & 0 & 0 \\ 0 & 0 & \beta|\dot{x}_2 - \dot{x}_1| + \frac{\gamma(\dot{x}_2 - \dot{x}_1)}{|z|} \end{bmatrix} \tag{9}$$

$$C_{nd} = \frac{\partial K_{n(i)}}{\partial \dot{X}_{(i)}} = \begin{bmatrix} \frac{\partial K_{n(1)}}{\partial \dot{x}_1} & \frac{\partial K_{n(1)}}{\partial \dot{x}_2} & \frac{\partial K_{n(1)}}{\partial \dot{z}} \\ \frac{\partial K_{n(2)}}{\partial \dot{x}_1} & \frac{\partial K_{n(2)}}{\partial \dot{x}_2} & \frac{\partial K_{n(2)}}{\partial \dot{z}} \\ \frac{\partial K_{n(3)}}{\partial \dot{x}_1} & \frac{\partial K_{n(3)}}{\partial \dot{x}_2} & \frac{\partial K_{n(3)}}{\partial \dot{z}} \end{bmatrix} = \begin{bmatrix} 0 & 0 & 0 \\ 0 & 0 & 0 \\ \frac{-\beta(\dot{x}_2 - \dot{x}_1)}{|\dot{x}_2 - \dot{x}_1|z} - \gamma|z| & \frac{\beta(\dot{x}_2 - \dot{x}_1)}{|\dot{x}_2 - \dot{x}_1|z} + \gamma|z| & 0 \end{bmatrix} \tag{10}$$

Suppose the steady-state solution of the system is

$$x_{i0} = \sum_{k=1}^{n} a_{ik}\cos(2k-1)\tau_1 + \sum_{k=1}^{n} b_{ik}\sin(2k-1)\tau_1 = DA_i$$
$$\Delta x_{i0} = \sum_{k=1}^{n} \Delta a_{ik}\cos(2k-1)\tau_1 + \sum_{k=1}^{n} \Delta b_{ik}\sin(2k-1)\tau_1 = D\Delta A_i \qquad (i = 1, 2, 3) \tag{11}$$

among which

$$D = (\cos\tau_1, \cos 3\tau_1, \ldots, \cos(2n-1)\tau_1, \sin\tau_1, \sin 3\tau_1, \ldots, \sin(2n-1)\tau_1)$$
$$A_i = (a_{i1}, a_{i2}, \ldots a_{in}, b_{i1}, b_{i2}, \ldots b_{in})^T \tag{12}$$
$$\Delta A_i = (\Delta a_{i1}, \Delta a_{i2}, \ldots \Delta a_{in}, \Delta b_{i1}, \Delta b_{i2}, \ldots \Delta b_{in})^T$$

The steady-state solution of the system is equivalent to the following Equation (13):

$$x = SA, \delta x = S\delta A \tag{13}$$

among which

$$A = \begin{bmatrix} A_1^T & A_2^T \end{bmatrix}^T, \Delta A = \begin{bmatrix} \Delta A_1^T & \Delta A_2^T \end{bmatrix}^T$$
$$S = \begin{bmatrix} D & \\ & D \end{bmatrix} \tag{14}$$

Substitute Equation (13) into Equation (8), and apply the Galerkin algorithm on both sides of the equation to obtain

$$\int_0^{2\pi} \delta x^T \left[ \omega_0{}^2 M \Delta \ddot{x} + \omega_0 C \Delta \dot{x} + K \Delta x + K_{nd} \Delta x + C_{nd} \Delta \dot{x} \right] d\tau = \\ \int_0^{2\pi} \delta x^T \left\{ \left[ \Phi - (2\omega_0 M \ddot{x}_0 + C \dot{x}_0 + K_n(\dot{x}_0)) \right] \right\} \Delta \omega d\tau \tag{15}$$

Substitute Equation (7) into the formula:

$$\int_0^{2\pi} S^T \left[ \omega_0{}^2 M \ddot{S} + \omega_0 C \dot{S} + KS + K_{nd} S + \omega_0 C_{nd} \dot{S} \right] d\tau \Delta A = \int_0^{2\pi} \Phi d\tau - \\ \int_0^{2\pi} S^T \left[ 2\omega_0 M \ddot{S} + C \dot{S} + K_n \right] d\tau A \Delta \omega \tag{16}$$

Order:

$$K_{mc} = \int_0^{2\pi} S^T \left[ \omega_0{}^2 M \ddot{S} + \omega_0 C \dot{S} + KS + C_{nd} \dot{S} + \omega_0 K_{nd} S \right] d\tau \\ R_{mc} = \int_0^{2\pi} S^T \left[ 2\omega_0 M \ddot{S} + C \dot{S} + K_n \right] d\tau \\ R = \int_0^{2\pi} \Phi d\tau \tag{17}$$

The linear equations can be obtained as follows:

$$K_{mc} \Delta A + R_{mc} A \Delta \omega = R \tag{18}$$

When studying the relationship between the vibration response of each degree of freedom of the system and the change in excitation frequency, $\Delta \omega$ can be made to be the active increment, where $\Delta \omega = 0$ in each solution point, and thus Equation (17) becomes

$$\Delta A = R / K_{mc} \tag{19}$$

In the iterative process, firstly, a set of initial test values of the steady-state response under the corresponding external load can be calculated. Then, the remainder is obtained according to the above equation. If the corresponding threshold is not reached, we can continue to apply a small increment to the excitation frequency, as $\omega^{(i+1)} = \omega^{(i)} + \Delta \omega$, and then proceed to the next iteration. If $R$ exceeds the corresponding threshold, iteration Equation (20) shall be continued until it reaches the allowable range, and then the amplitude–frequency characteristic curve of the structure can be obtained.

$$K_{mc}{}^i \Delta A^{(i+1)} = R^i, \ A^{(i+1)} = A^i + \Delta A^{(i+1)} \tag{20}$$

## 4. Analysis of the Influence Factors of the Nonlinear Isolation System

### 4.1. Influence of the Isolation Control Parameters

For the isolation structure using the Bouc–Wen model, $\beta$ and $\gamma$ can effectively reflect the dynamic characteristics of the isolation system. Therefore, the steady-state responses of each particle of the system under different control parameters were analyzed in the two cases, as shown in Table 1.

**Table 1.** Value table of the model control parameters.

| Parameters | | $m_1$ | $m_2$ | $c_1$ | $c_2$ | $k_1$ | $k_2$ |
|---|---|---|---|---|---|---|---|
| **Values** | | 1 | 8 | 0.2 | 0.4 | 4 | 2 |
| $\gamma \geq 0$ | $\beta$ | | | 0.25 | 0.5 | 0.75 | 1.0 |
| | $\gamma$ | | | 0.75 | 0.5 | 0.25 | 0 |
| $\gamma \leq 0$ | $\beta$ | | | 0.1 | 0.25 | 0.5 | 0.75 |
| | $\gamma$ | | | −0.9 | −0.75 | −0.5 | −0.25 |

Figure 4a shows that when $\gamma \geq 0$, all particles of the isolation system drift to the left, which shows the soft mechanical properties. With the increase in the ratio of the

control parameters $\beta/\gamma$, the amplitude response of each particle of the structural system becomes smaller, which proves that the energy dissipation effect of the structure is better and the energy transmission characteristics are more complete under this parameter. By comprehensive comparison with Figure 4, it can be found that in the isolation system of Bouc–Wen model with this parameter, there is no unstable solution in the first- and second-order resonance region of each particle, and there is no amplitude jump phenomenon in the peak curve. The whole system is stable.

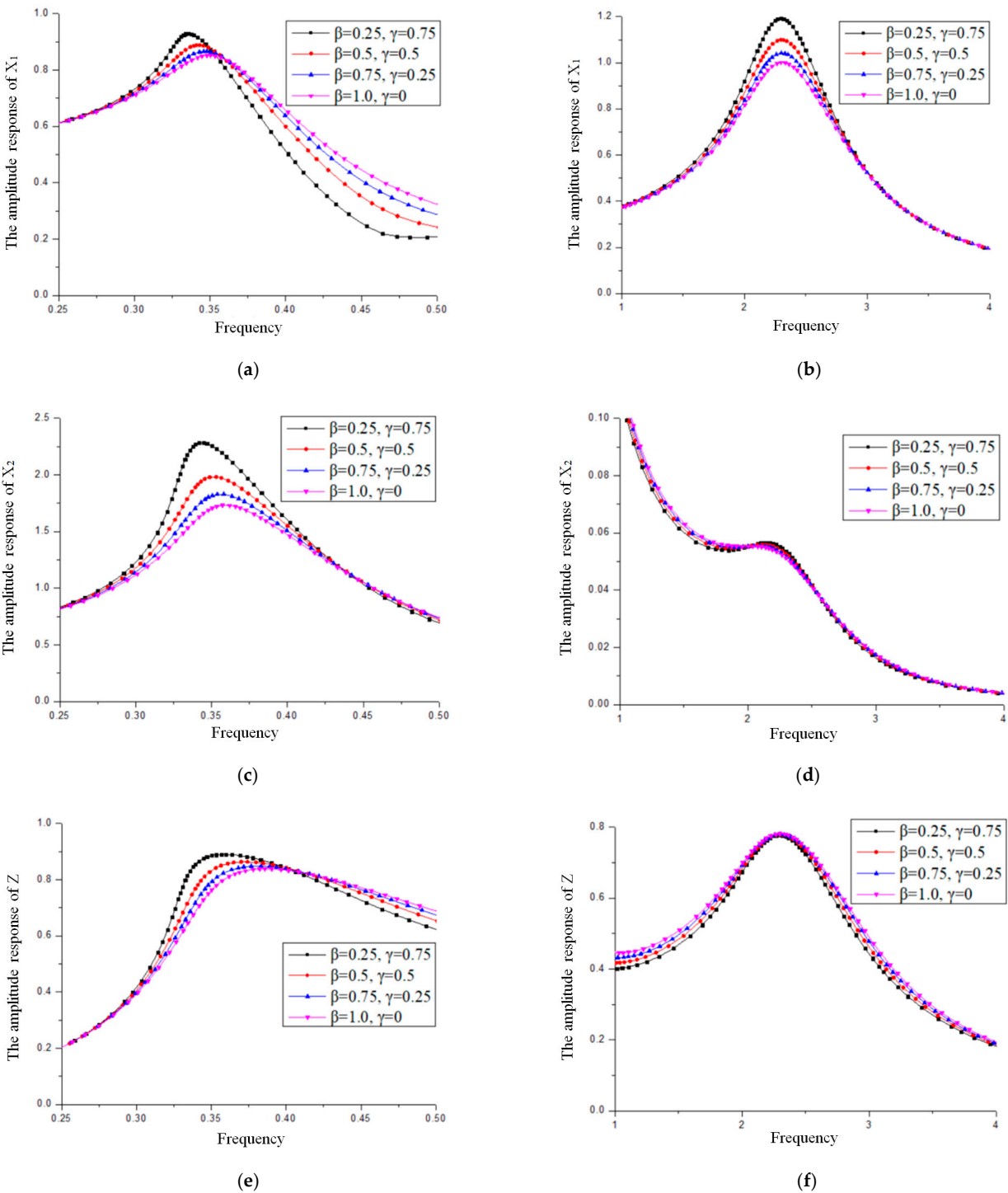

**Figure 4.** Amplitude–frequency characteristic curve of the isolation structure under different control parameters ($\gamma \geq 0$) (**a**,**b**) Amplitude–frequency characteristic curve of the substructure; (**c**,**d**) Amplitude–frequency characteristic curve of the superstructure; (**e**,**f**) Amplitude–frequency characteristic curve of the hysteretic element.

Figure 5 shows that when $\gamma < 0$, all particles of the isolation system drift to the right, which shows the hardened mechanical properties. At this time, the material characteristics' change will make the system particles appear as having an unstable solution with jumping phenomena. By comparing Figure 5a–c, it can be found that as the ratio of the control parameters $\beta/|\gamma|$ decreases, the amplitude response of each particle of the structural system gradually increases. In this case, the material in the isolation system appears as aging or hardening characteristics, which is easy to lead to the overall destruction of the isolation structure.

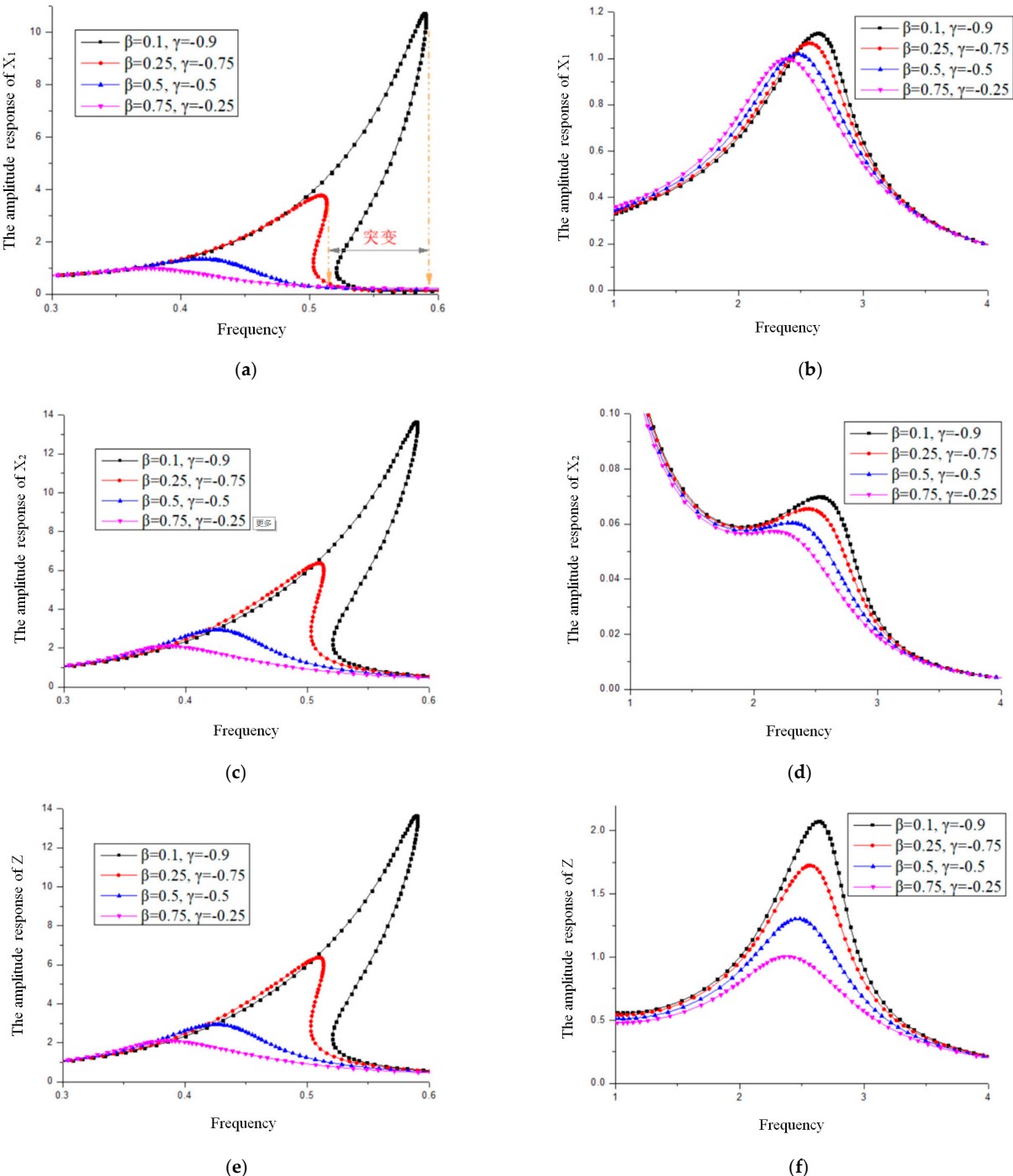

**Figure 5.** Amplitude–frequency characteristic curve of the isolation structure under different control parameters ($\gamma \leq 0$) (**a**,**b**) Amplitude–frequency characteristic curve of the substructure; (**c**,**d**) Amplitude–frequency characteristic curve of the superstructure; (**e**,**f**) Amplitude–frequency characteristic curve of the hysteretic element.

By comprehensive comparison of Figures 4 and 5, it can be found that when $\gamma \leq 0$, the isolation bearing appears as aging, corrosion, low-temperature hardening and other phenomena [36,37]. The steady-state response of each particle of the system has an unstable solution, and the peak curve of the first-order resonance region shows an amplitude jump phenomenon. In addition, it can be observed that if and only if $\beta/|\gamma|<1$, the structural system has an unstable solution and produces a large response. Therefore, the influence of the control parameter should be fully considered in the actual design.

### 4.2. Influence of the Damping Ratio of the Isolation Layer

When other control parameters are kept constant, the influence of the damping ratio on the isolation system can not be underestimated. As shown in Figure 6, when the damping ratio of the isolation layer is relatively small, a large amplitude jump occurs in the range of the first-order resonance zone of each particle. With the increase in the damping ratio, the unstable solutions of each particle gradually disappear, and the system response gradually presents a stable state. As can be seen from Figure 6a–c, the response of the system does not decrease with the damping ratio, which gradually increases when it is larger than the critical value. Therefore, the reasonable selection and placement of damping materials and damping elements can make the material properties work better, and it is necessary to consider the optimal selection of different damping ratios for the target model in the actual design.

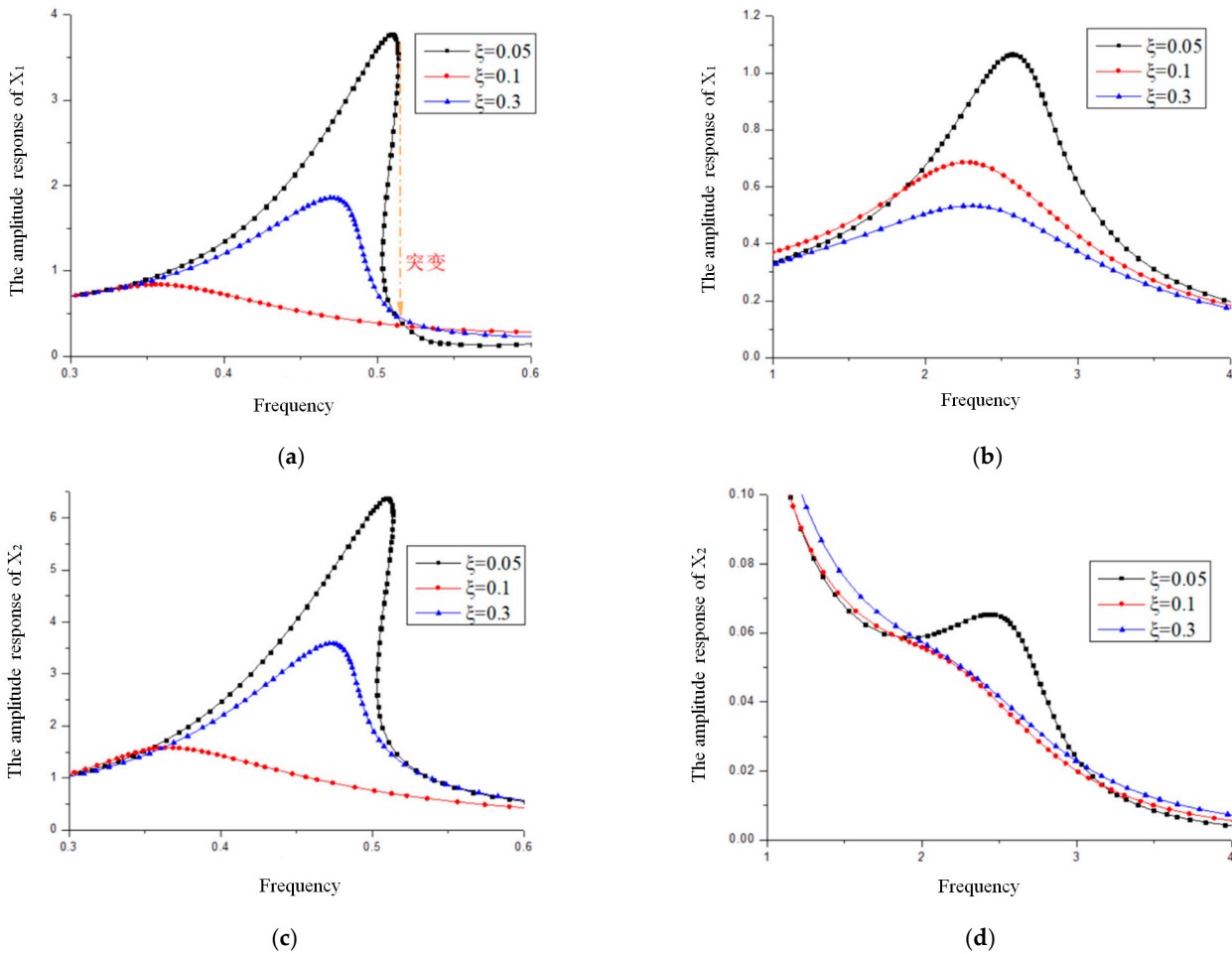

**Figure 6.** *Cont.*

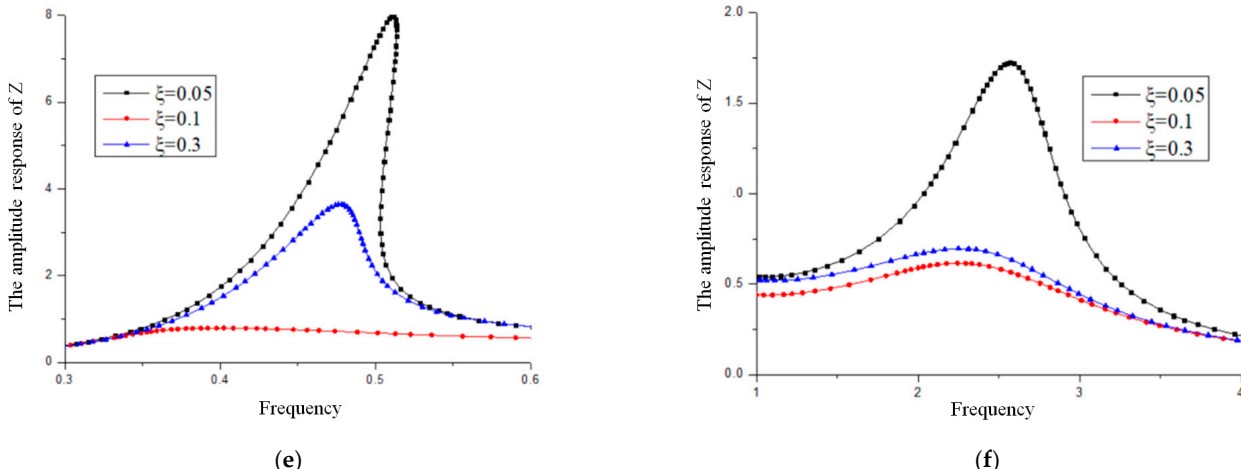

**(e)**          **(f)**

**Figure 6.** Amplitude–frequency characteristic curves of the isolation structures with different damping ratios (when $\xi = 0.05$, 0.10.3, $\beta = 0.25$, $\gamma = -0.75$) (**a**,**b**) Amplitude–frequency characteristic curve of the substructure; (**c**,**d**) Amplitude–frequency characteristic curve of the superstructure; (**e**,**f**) Amplitude–frequency characteristic curve of the hysteretic element.

### 4.3. The Influence of External Excitation Amplitude

In order to verify the accuracy and effectiveness of the incremental harmonic balance method, the Runge–Kutta method is used in this section to jointly verify the stability of the isolation system under the influence of the parameters.

Equation (4) is transformed into an autonomous system as follows (Equation (21)):

$$\begin{cases} \dot{x}_1 = x_4 \\ \dot{x}_2 = x_5 \\ \dot{x}_3 = (x_5 - x_4 - \beta(x_5 - x_4)x_3 - \gamma(x_5 - x_4)|x_3|) \\ \dot{x}_4 = m_1^{-1}(F\cos(\omega t) - k_1 x_1 - k_2 x_3(\alpha - 1) - c_1 x_4 - \alpha k_2(x_1 - x_2) - c_2(x_4 - x_5)) \\ \dot{x}_5 = m_2^{-1}(c_2(x_4 - x_5) + k_2 x_3(\alpha - 1) + \alpha k_2(x_1 - x_2)) \end{cases} \quad . \tag{21}$$

As shown in Figure 7, the Runge–Kutta method and the IHB analysis results are almost consistent, which proves the accuracy of the results. With the increase in the amplitude of the external excitation, the superstructure and the substructure in the first-order resonance region have different degrees of jump phenomena, which can present an unstable state, and the second-order resonance region of the peak is relatively small.

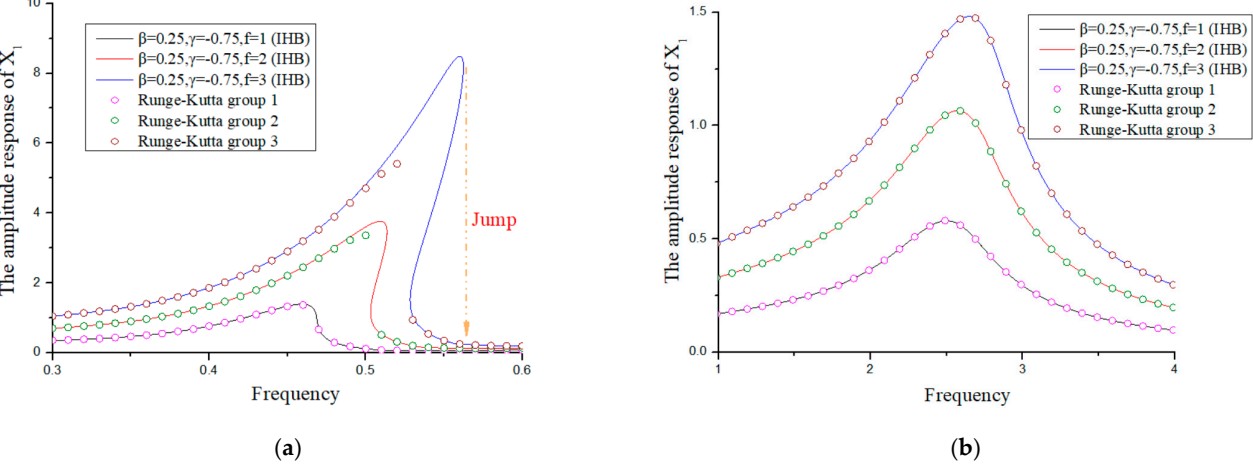

**(a)**          **(b)**

**Figure 7.** *Cont.*

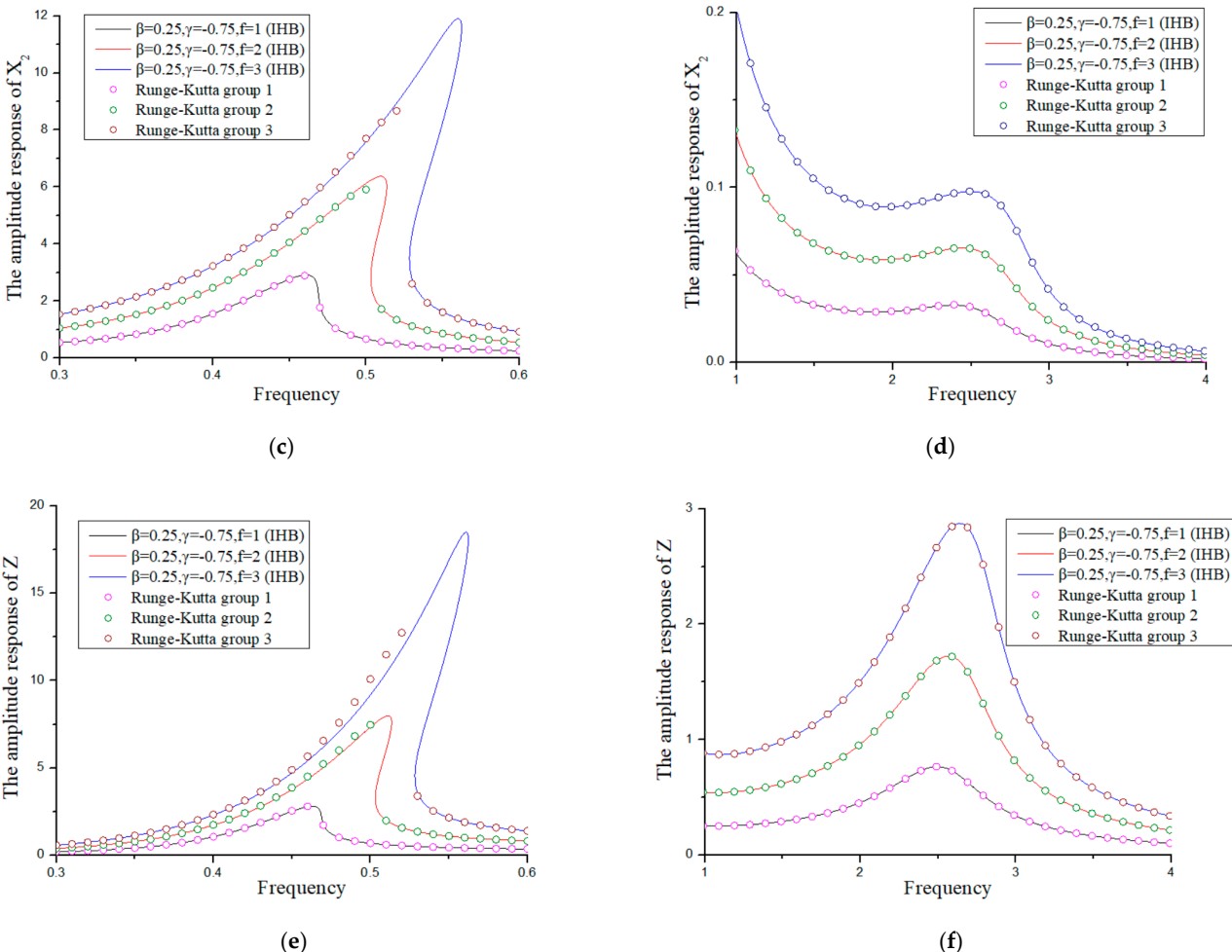

**Figure 7.** Amplitude–frequency characteristic curve of the isolation structure under different excitation amplitudes (when $\beta = 0.25$, $\gamma = -0.75$) (**a,b**) Amplitude–frequency characteristic curve of the substructure; (**c,d**) Amplitude–frequency characteristic curve of the superstructure; (**e,f**) Amplitude–frequency characteristic curve of the hysteretic element.

### 4.4. Influence of Mass Ratio

Five different mass ratios are selected in this section, which are as follows: $\mu = 8:1$, $\mu = 4:1$, $\mu = 2:1$, $\mu = 1:1$, and $\mu = 0.5:1$. The mass ratio is the ratio of the mass of the superstructure to that of the substructure; i.e., $\mu = m_2/m_1$, which can represent the different layers and forms of the actual structure.

Figure 8 shows the amplitude–frequency characteristic curve of each particle in the system as the mass changes; it was found that when the mass ratio of the structure is $\mu = 8:1$, there will be an obvious jump phenomenon in the first-order resonance region of each particle, and the system is in an unstable state. As the mass ratio decreases, the unstable region gradually disappears, and the influence factor has a significant weakening effect on the response amplitude. Meanwhile, the peak value gradually moves from the low frequency region to the high frequency region. However, when the mass ratio decreases to less than 1:1, as the mass ratio of the superstructure is small, the isolation system will have a larger amplitude response.

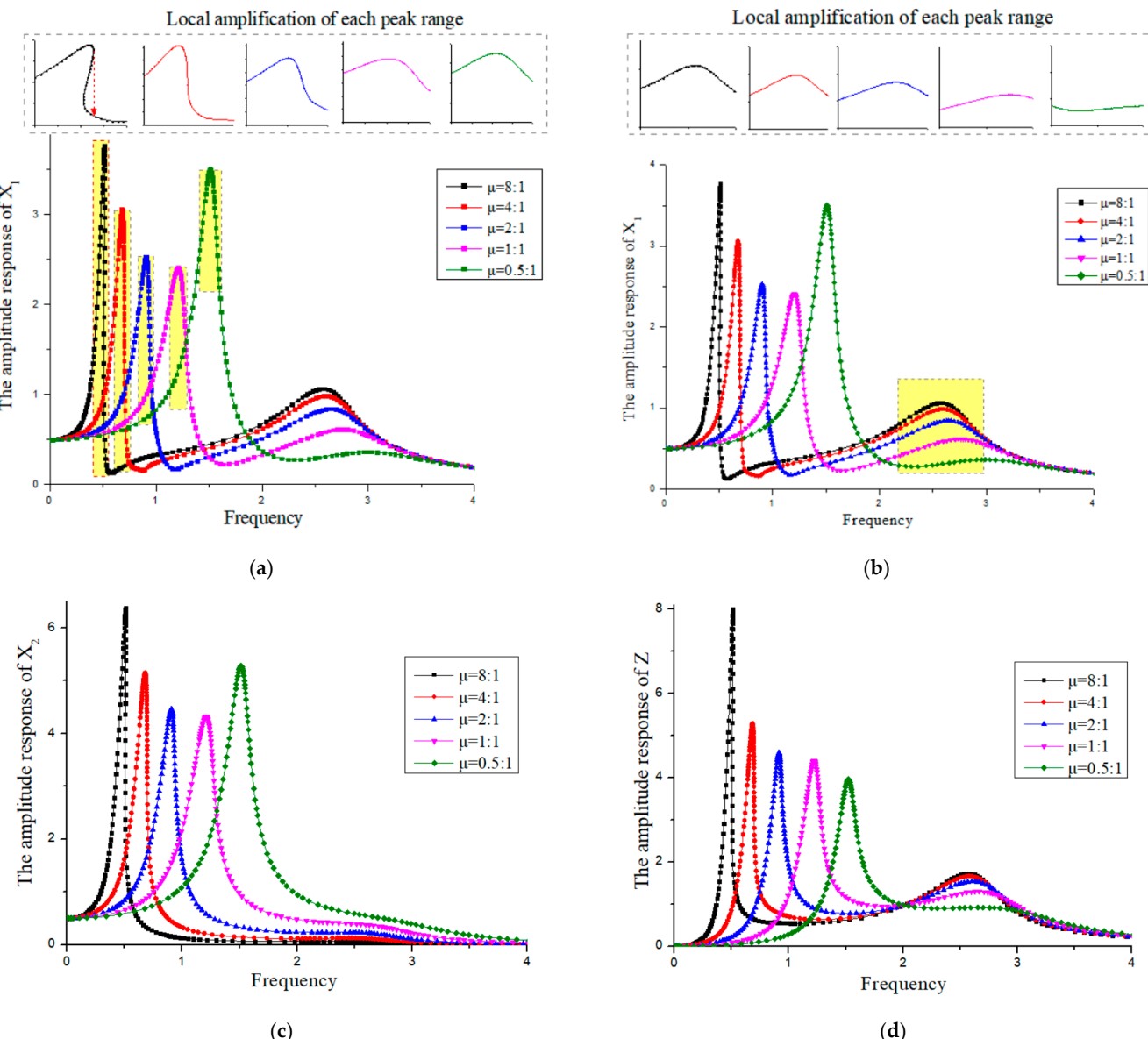

**Figure 8.** Amplitude–frequency characteristic curves of the isolation structures with different mass ratios (**a**,**b**) Amplitude–frequency characteristic curve of the substructure; (**c**) Amplitude–frequency characteristic curve of the superstructure; (**d**) Amplitude–frequency characteristic curve of the hysteretic element.

## 5. Conclusions

The incremental harmonic balance method was used to solve the Bouc–Wen isolation system under different influence factors. Furthermore, a stability analysis was carried out, and the accuracy of the results verified by the Runge–Kutta method. The influence of the isolation control parameters, damping ratio and external excitation amplitude parameters on the isolation system was obtained, which provides a basis for practical engineering designs.

**Author Contributions:** Conceptualization, X.T.; methodology, Z.Z.; software, X.T.; validation, X.G., Z.Z. and X.T.; formal analysis, X.T.; investigation, X.G.; resources, Z.Z.; data curation, Z.Z.; writing—original draft preparation, X.G.; writing—review and editing, X.T.; visualization, X.G.; supervision, Z.Z.; project administration, Z.Z.; funding acquisition, Z.Z. All authors have read and agreed to the published version of the manuscript.

**Funding:** The authors wish to acknowledge the financial support by National Natural Science Foundation of China (Grant No. 51478387).

**Institutional Review Board Statement:** This study did not involve humans or animals.

**Informed Consent Statement:** This study did not involve humans.

**Data Availability Statement:** This study did not report any data.

**Conflicts of Interest:** The authors declare no conflict of interest.

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
