# Peer review of "Dynamic Characteristics of the Bouc–Wen Nonlinear Isolation System"

_applsci, doi:10.3390/app11136106_

Round 1

Reviewer 1 Report

Dear Authors,

your manuscript is quite interesting and could deserve to be published in Applied Sciences with the minor corrections/ adjustments suggested in my report. 

Kind regards

Author Response

Deer reviewer,

I'm sorry for the late reply to the revised draft.The modifications are as follows, and uploaded the revised full text.

Reply,

1.The introduction expands on the literature which provided by the reviewer.

2. The isolation structure is simplified into two degrees of freedom,and gave an explanation.

3. In the 4 section, the values of relevant parameters are given in the Table 1.

4. The definition of mass ratio is added.

Best Wishes.

Reviewer 2 Report

Dear Authors,

I have read your paper with attention and pleasure.

In my opinion, the manuscript Dynamic Characteristics of Bouc-Wen Nonlinear Isolation System  presents original research and could be interesting for readers of the Applied Sciences Journal. The motivation is clear. The object of study, as well as the results, are comprehensively described providing valuable conclusions.

The paper is organised in a logical manner. The state of art covers the main results in the field, including the authors’ own results. The contributions of the paper are clearly stated in the Introduction chapter.

I have no objections to recommend publishing this paper. However, due to the listed below drawbacks, my recommendation is " Accept minor revision". In my opinion, several aspects require clarification. Please revise and add some comments and improvements according to the following:

  • page 3, statement: " the ratio of β/ γ is relatively large" - please  specify what "relatively large" means, from the values given in Fig. 2 it appears that this ratio is 1 similarly with the statement is "relativly small" - to what value of the coefficient do you assume that it is a small value 
  • Please try to define more broadly the practical meaning of the obtained  the  influence law  of isolation  control  parameters,  damping  ratio  and  external  excitation amplitude   parameters   on   the   isolation   system .

editorial notes:

- page 3: Fig.  2(a,c)is - space missed

- Fig. 4-7 -  the caption for subfiger a and b is imprecise 

general note - the math formulas and Greek characters look too large for the text of the article 

Author Response

Deer reviewer,

I'm sorry for the late reply to the revised draft.The modifications are as follows, and uploaded the revised full text.

Reply,

  1. The ratio of β/IγI represents the area of the hysteretic loop in the figure.
  2. In the 4 section, the values of relevant parameters are given in the Table.
  3. The format of the full text has been modified.

Best Wishes.
